# Feasibility and acceptability of research-grade wearables for health and labor capacity monitoring in the context of climate change and heat stress: The case of Nouna, Burkina Faso

Georgi Zout[1]*, Celine Höver[1], Edgar Eggert[1], Hanns-Christian Gunga[1], Lucienne Ouermi[2], Ali Sié[2,3], Sophie Huhn[3], Aditi Bunker[3], Sandra Barteit[3], Stefan Mendt[1‡], Martina Anna Maggioni[1,4,5‡]

1 Charité – Universitätsmedizin Berlin, Institute of Physiology, Center for Space Medicine and Extreme Environments Berlin, Berlin, Germany, 2 Centre de Recherche en Santé de Nouna (CRSN), Burkina Faso, 3 Faculty of Medicine and University Hospital, Heidelberg Institute of Global Health (HIGH), Heidelberg University, Heidelberg, Germany, 4 Department of Biomedical Sciences for Health, Università degli Studi di Milano, Milano, Italy, 5 Charité, Universitätsmedizin Berlin, Charité Center for Global Health (CCGH), Berlin, Germany

‡ SM and MAM are joint senior authors on this work.
* georgi.zout@charite.de

## Abstract

Current climate projections estimate a further rise of mean ambient temperatures of 1.5°C until 2040. However, the understanding of heat stress's impact on health and labor capacity, especially in vulnerable regions such as sub-Saharan Africa, remains limited. In turn, no long-term investigations monitoring and uniting both individual-level subject and environmental data have been yet conducted in this region. To address this knowledge gap, we evaluated the feasibility and acceptability of research-grade wearables for continuous, direct, individual-level monitoring of physiological parameters in a population of subsistence farmers (one woman and one man per n=20 households) in rural Burkina Faso. We conducted a four-week pilot study, investigating data completeness and quality of heart rate and core body temperature, and data completeness for physical activity, and GPS individual tracking, simultaneously monitoring outdoor and indoor wet-bulb-globe temperature. Additionally, participants were surveyed regarding their acceptance of employed wearables. Regarding environmental indoor monitoring, we collected 85% of completed data, whereas for outdoor, it was 100%. An average of 97.5% of viable data sets were retrieved for all wearables (heart rate: 97.5%, core body temperature: 97.5%, physical activity: 97.5%, GPS: 97.5%). Individual data point completeness was > 92% for all sensors, except GPS, where it was 67% on average. Acceptance of wearables was positive, with a range of 79% to 95%. The main challenges perceived by participants were missing personalized sensor feedback (70%) and uncertainty regarding the meaning of the wearables (47.5%). We show that the implementation

**Data availability statement:** All relevant data are within the paper and its Supporting Information files.

**Funding:** This study was supported by the Deutsche Forschungsgemeinschaft (DFG) as part of the FOR 2936 "Climate Change and Health in Sub-Saharan Africa". The funders had no role in study design, data collection and analysis, decision to publish, or preparation of the manuscript. The study has been approved by DFG (FOR 2936/grant number: 414/6–1/660477). The recipient of the grant was Prof. Hanns-Christian Gunga. We also acknowledge the support from the Open Access Publication Fund of Charité – Universitätsmedizin Berlin for publication costs.

**Competing interests:** The authors have declared that no competing interests exist.

of research-grade wearables in sub-Saharan Africa is technically feasible and socially accepted. Further, we point out current challenges and provide a solid framework for future research in this region.

## Introduction

According to the 2023 IPCC Synthesis Report, global warming has reached 1.1°C relative to pre-industrial levels, with a further rise of mean ambient temperatures of approximately 1.5°C likely to occur until 2040 [1]. This development predicts more frequent and severe extreme weather events, the destruction of ecosystems, and detrimental impacts on human health and societies. Moreover, higher environmental temperatures translate to heightened rates of heat stress in humans. Inducing an increase in core body temperature (CBT), heat stress challenges the body's heat dissipative mechanisms to uphold thermal homeostasis, such as skin and organ perfusion regulation [2]. The impairment of physiological heat dissipation causes heat strain, generating various medical hazards, i.e., dehydration, cardiovascular strain, and heat stroke. Furthermore, heat stress may reduce physical work capacity and, thus, labor capacity significantly [3]. To objectively quantify and predict the effects of heat stress in humans, several environmental-based indices have been developed. One of the most used indices is the wet-bulb-globe temperature (WBGT) [4]. WBGT is an index that integrates air temperature, humidity, wind speed, and radiation. It is considered a standardized reference to assess the risk of heat-induced illness during physical activity or manual labor while exposed to the environment, providing a gauge of occupational health risks in hot environments [4,5]. Thus, it has been shown that even mild heat stress (18°C WBGT) may cause a 10% reduction in physical work capacity, with most extreme heat conditions (40°C WBGT) potentially reducing it even up to 78% [6].

Tropical and desert climate zones, such as sub-Saharan Africa (SSA), are exceptionally affected by climate change. There, intensified heat waves, droughts, heavy precipitation, and flooding are already observed [1]. Research has shown heat stress to be a particularly acute issue in the population of SSA, where residents experience a heightened risk of heat stress-related illness, with higher rates of heat strain-induced morbidity and mortality [1,7]. Due to increasing heat exposure, outdoor workers in SSA, especially in subsistence farming, face exacerbated occupational health risks and reduced productivity, with estimates of up to 50% reduction in physical work capacity throughout all seasons until 2030 [8,9]. Especially in low-resource, rural settings of SSA, subsistence farming is highly prevalent; for example, up to 86% of the population in Burkina Faso still relies on subsistence agriculture, according to USAID [10]. Forced to work during the daytime, with agricultural activities frequently taking place during the hottest months of the year, subsistence farmers are most vulnerable to extreme heat. Therefore, when investigating heat stress in SSA, considering and evaluating the potential loss of physical work capacity in subsistence farmers is crucial, as productivity losses pose a direct risk of food scarcity and malnutrition

[9–11]. The urgent need for adaptation strategies emphasizes the importance of better understanding the interrelation effect of intensified heat and occupational health risks, with a focus on labor capacity and the environmental well-being of vulnerable populations [2,9].

While WBGT portrays the environmental scope of heat stress in humans, there are several variables for directly quantifying physiological heat strain in the context of heat stress. As such, changes in CBT, heart rate (HR), and physical activity (ACT), can be utilized as either a direct reflection of or to predict physiological strain and adverse health outcomes [5,12–14]. Previously developed models incorporated those variables, notably HR and CBT, to assess and predict physiological and heat strain in naturalistic environments. Such models, referred to as physiological strain indices (PSI) have been developed to evaluate heat stress more precisely and further the standardization of such evaluation [15]. By exploring feasible set-ups for long-term, real-life data collection, future refinement and adaptation of the PSI may be possible.

With the rise of wearable technologies, decentralized health monitoring and advances in heat stress and strain assessment, a promising opportunity to investigate vulnerable groups lies in employing these wearable devices (wearables) [5,16]. The technological developments of sophisticated wearables have led to frequent use in clinical medicine and research [17]. Historically, research on heat stress in hot and desert climate zones in low-resource contexts relied primarily on hospital, meteorological, and survey data [18,19]. However, significant technological advancements in the field of consumer-based wearables have enabled the implementation of such technology in rural contexts of SSA [20,21]. However, to our knowledge, no previously published research has used research-grade wearables to continuously monitor subjects in a naturalistic SSA setting. Here, an interplay of technological, systemic, and human factors influences the successful implementation and integration of wearables, especially in low-resource contexts [22]. Therefore, a feasibility and acceptability pilot study is helpful to examine the realizability of such a novel approach [23].

With a multitude of negative health impacts and an especially severe affection of vulnerable groups in hot and desert climate zones, such as rural Burkina Faso, heat stress poses a serious danger to millions of people. This is further exacerbated by the intertwinement of occupational health risks brought on by heat stress and the stark reliance on subsistence agriculture and thus manual labor in said region. To reduce risk and develop sophisticated adaptational strategies for SSA inhabitants, particularly rural population groups, such as subsistence farmers, an in-depth understanding of the underlying mechanisms is urgently required. To accomplish this overarching goal, in a first step, we aim to close the gap between the continuous assessment of physiological responses through research-established wearables and the evaluation of climate change's impact on the inhabitants of SSA regarding their health, physical work capacity, and occupational risks. To lay the groundwork, we conducted a feasibility and acceptability study in the low-resource, rural setting of Nouna, Burkina Faso, among a population group of subsistence farmers. This pilot study had two aims: (I) to investigate the technical feasibility of research-grade wearables for quantifying the physiological variables HR, CBT, and ACT, the environmental parameter WBGT, and the participants' geo-location in GPS coordinates in the given setting; (II) to assess the acceptability of the devices, as well as of the experimental protocol and procedures, among the study participants. Both aims were achieved by evaluating data completeness and quality and the participant surveys.

## Methods

### Study setting and recruitment

The sample was recruited within the Health and Demographic Surveillance System (HDSS) of Nouna, Burkina Faso, which has a population of about 125,000. This surveillance system continuously monitors a dynamic cohort of the total population within a defined area [24]. The Nouna HDSS is located at a low altitude of about 227 m above sea level with a sub-Saharan climate profile and a rainy season from May to September [25]. Our specific sample was derived from the sample planning of our co-project conducted by Huhn et al [21]. In this context, n = 7 villages nearest an assigned

healthcare facility were randomly selected. Our main criteria during the initial recruitment process were an age range of 18–55 and the fact that participants had to reside within 10 km of the Nouna weather station that we had previously installed. For enrollment, our field workers went to potential participants' households, informed them of the study setup, procedures, and duration, and thus obtained informed consent if participation was feasible. Additional details on the study sampling procedure and the recruitment specifications are given elsewhere [21,26].

The final study sample comprises 20 men and 20 women (N = 40). Inclusion criteria for participants were: (I) age between 18 and 55 years, (II) active farmers, men and women belonging to the same household (household refers to couples sharing the same place of residence), (III) residing within 10 km radius of the nearest weather station and having no plans to move during the duration of the study, (IV) no cardiovascular, metabolic, neurodegenerative or orthopedic impairments, (V) no alcohol or drugs abuse and (VI) not being pregnant, breastfeeding or having the intention of getting pregnant within study duration. The rationale for including men and women sharing a household was to increase protocol adherence, ensure lower drop-out rates and streamline sample planning as well as recruitment procedures. Further, including both men and women enabled us to evaluate the feasibility of our approach more thoroughly including a gender-specific perspective.

## Experimental procedure

We conducted two study runs of two weeks' duration, with a break of about one week in between (February-March 2021). Each run incorporated n = 20 (ten women, ten men). The recruitment period for the first run was from February 10 to February 11, 2021 and for the second run March 6 to March 8, 2021. There were a total of three contacts with participants during each respective two-week run. After our field personnel gave participants a full explanation of the procedures to be conducted, the participants received a written copy and gave informed consent in written form. An entry survey to further check eligibility was conducted. (S1 Entry Survey). The study and all accompanying procedures and preliminary work were approved by the Comité d'ethique pour la recherche en santé (approval date: Ocotber 7, 2020; 2020-10-2016) in Burkina Faso and by the ethical committee of the Charité – Universitätsmedizin Berlin, Germany (approval date: March 11, 2019; EA1/060/19). During the first contact, participants were given a wrist-worn accelerometric tracker. A portable hygrometer for WBGT assessment was placed on the indoor wall by trained personnel. Both devices were set to record data continuously for 14 days. The second contact occurred, when participants were asked to return to the assigned healthcare facility on the 12th day. They were subsequently equipped with multiple wearables recording HR, CBT, and GPS tracking for up to 24 hours. Additionally, participants underwent a medical check-up, including an assessment of anthropometric data., such as age in years (yrs), height (h) in centimeters (cm), weight (w) in kilograms (kg), and blood pressure SBP (systolic) and DBP (diastolic) in mmHg and a bioimpedance analysis (BIA). After completing the two weeks run participants returned to the healthcare facility for a third contact, returned the handed-out sensors and underwent an acceptability questionnaire. Fig 1 depicts the flow of study and experimental procedures.

## Environmental monitoring

Because environmental factors can affect human health and performance, we collected data on both the indoor and outdoor environment. Outdoor environmental data were recorded at 15-minute intervals by a local weather station equipped with ADCON sensors (OTT Hydromet GmbH, Germany). Real-time data was accessed and downloaded via a dedicated server. Since the weather station is not equipped with a black globe, it only provides estimates of the outdoor WBGT. According to Carter et al., this is achieved by incorporating the natural wet bulb temperature ($T_{nwb}$), solar radiation (SR), relative humidity (RH), and air temperature ($T_a$) [27]:

$$WBGT = 0.7 \times T_{nwb} + 0.2 \times [0.009624 \times SR + 1.102 \times T_a - 0.00404 \times RH - 2.2776] + 0.1 \times T_a$$

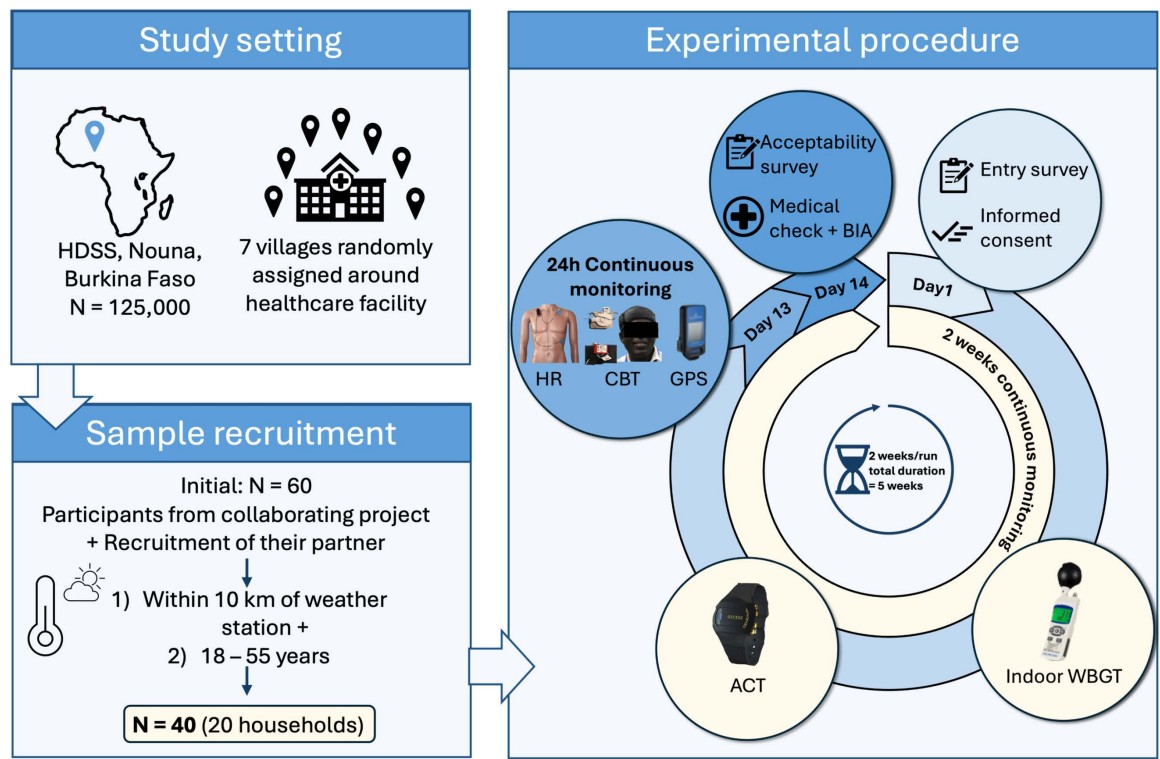

**Fig 1. Study design and flow of study procedures.**

Indoor environmental data were recorded with the portable hygrometer PCE-WB 20SD (PCE Deutschland GmbH, Germany), placed on the indoor wall within the room where study participants spent most of their indoor time and at a height of about 170 cm to be out of reach for children. In addition to air temperature and relative humidity, this device estimates WBGT in accordance with the ISO 7243 specification for WBGT measurement without solar load: $WBGT = 0.7 \times T_{nwb} + 0.3 \times T_g$ where $T_{nwb}$ represents natural wet bulb temperature and $T_g$ represents the black globe temperature [28]. Data were recorded at 10-minute intervals on an integrated SD card.

### Physiological monitoring

GENEActiv (Activinsights Ltd., UK) is a validated, lightweight wrist-worn device (43 x 40 x 13 mm, weight: 28 g) that provides continuous recording of tri-axial acceleration (range ±8 $g$), near body temperature, and ambient light [29]. Post-processing of the GENEActiv data enables an objective assessment of energy expenditure, physical behavior, sleep behavior, circadian rhythm, and step count [30–34]. GENEActiv was configured to collect data at a frequency of 10 Hz.

The participants' core body temperature (CBT) was continuously estimated using a mobile and invasive monitoring system, a Tcore™ sensor (Drägerwerk AG & Co. KGaA, Lübeck, Germany) placed on the forehead and connected to a data logger (HealthLabFunkMaster, KORA Industrie-Elektronik GmbH, Hambühren). We integrated both components into a custom-made headband which allowed the study participants to pursue their activities while the temperature was recorded. The sensor technology has been found to estimated CBT accurately in clinical and non-clinical scenarios [35–38].

To derive HR, study participants were equipped with the Faros Bittium 180™ (Bittium Biosignals Ltd, Finland), a gold standard ambulatory electrocardiography monitor. This small and lightweighted device (dimensions: 48 x 29 x 12 mm, weight: 13 g) enables precise long-period beat-to-beat measurements [39–41]. The sampling rate was set at 250 Hz.

The spatial movements of the study participants were tracked using a wearable GPS logger (GP-102 G-Porter GPS Logger, CanMore Corporation, Taiwan). Speed, altitude, and geo-location were recorded at 1-minute intervals on an internal storage. The device was connected to a strap and worn around the neck.

Further technical details can be found in a previously published study protocol by Barteit et al. [26].

## Survey on acceptability

We developed a questionnaire to explore the acceptability of wearables among study participants. The development of the questionnaire was done jointly with co-project researchers and our local study coordinators to accommodate to local specifications. The work was rooted in previous experience and the knowledge of the local community. Our questionnaire was originally designed in English. However, as our participants' native language is French, the local study coordinator translated the questionnaire to French prior to administration (S3 Acceptability Questionnaire). Survey Solutions, a survey software developed by the World Bank Group (Washington, DC 20433 USA), was used to design and conduct the survey [42,43].

The questionnaire consisted of 5-item Likert-scale single-select questions, multi-select, and open-ended questions. The acceptability questionnaire examined participants' acceptance and experiences of and with the respective wearables.

Upon finalizing the respective study run, participants returned to the assigned healthcare facility, where trained operators administered the questionnaire employing the computer-assisted personal interview method.

After administration and retrieval, we grouped the questionnaire data into thematic subsections, evaluating the personal reception, perceived practicability and value, social acceptance, and personal hindering factors for further participation. Through this approach, we covered major themes relevant to evaluating the acceptability of such a framework in potential long-term studies.

## Data analysis

We assess the feasibility of our approach by evaluating data completeness and, if possible, data quality. For this purpose, we processed the available recordings as follows: For HR, we used the Pan-Tompkins algorithm implemented in the *rpeaks* package for R-peak detection and the *RHRV* package for heart rate extraction [43,44]. Before collapsing in 1-minute epochs, HR below 40 bpm or above the predicted individual maximum were considered as artifacts and excluded. For CBT, temperatures outside the range of 36–40°C were identified as artifacts and then deleted. [45,46]. The temperatures were then collapsed in 1-minute epochs. For data sets in which HR was recorded together with CBT, we calculated the PSI. Since we could not determine resting values for HR and CBT in a thermoneutral environment, we did not use the initial values according to Moran et al. [15]. Instead, we calculated PSI using fixed resting values (CBTrest: 37°C, HRrest: 70 bpm), kept CBTcritical at 39.5°C, and set HRcritical to 90% of the individual maximum heart rate [47,48]. To determine the completeness of the GPS data, we checked whether the GPS device was continuously receiving a GPS signal (i.e., contain no time gaps in the records). GENEActiv files were imported in R using the *GENEAread* package [49]. The 10-Hz data were collapsed in 1-minute epochs. In addition, the standard deviation of acceleration in each axis (x-, y- and z-axis) and the sum of the signal vector magnitude (SVM) with gravity (*g*) subtracted were calculated for each 1-minute epoch.

$$\text{SVM} = \sum \left| \sqrt{x^2 + y^2 + z^2} - g \right|$$

To determine the potential data loss caused by not wearing GENEActiv, we used the approach of Barakat et al. [50], which considers the standard deviation of the accelerations and a drop in GENEActiv recorded near body temperature. For data sets recorded with the hygrometer (indoor data), measurement times were classified as artifacts and removed if the corresponding air temperature was greater than 50°C.

Acceptability has been previously defined by the technology acceptance model (TAM), which evaluates new technology's perceived usefulness and ease of use [51]. The TAM has been implemented to investigate the acceptance of consumer-grade and medical-grade wearable technologies, successfully identifying factors such as perceived benefit and perceived risk to be influential in the overall perception of a wearable [52]. To explore those dimensions, we formulated according statements, which assessed participants agreement or disagreement. The statements in our questionnaire were either presented as closed Yes-No questions, open-ended questions or Likert items. For Likert items the answers were coded numerically as follows: Strongly agree = 1, Agree = 2, Undecided = 3, Disagree = 4, Strongly Disagree = 5.

Descriptive statistic are presented as mean (M) and standard deviation (SD) unless stated otherwise. Differences between men and women on anthropometrics, on data completeness, and on the acceptability data were tested using the Wilcoxon rank-sum test [53]. The level of significance was set at 0.05 (two-sided). Statistical analysis and visualization were performed using R (version 4.3.3) in R Studio (version 2024.12.0.467) [54,55]

## Results

All recruited subjects (n = 20 women and n = 20 men) successfully participated in our study. Anthropometric and body composition data from the study sample are given in Table 1.

### Feasibility

We equipped the study participants with wearables for recording physiological data (24-hour recordings) and with data loggers for recording environmental factors (12-day recordings).

Fig 2 shows the data of one participant. Outdoor WBGT and air temperature were characterized by a greater daily fluctuation than indoors (Fig 2). For example, the outdoor air temperature fluctuated daily between 20 and 40°C, while the indoor air temperature, though at a higher level, only fluctuated between 30 and 37°C.

After the two study runs, 179 recordings were available from an expected 180 recordings. One HR recording was lost. Furthermore, six recordings had to be excluded from the evaluation (incorrect assignment: n = 2, lost timestamp: n = 2, short recording duration: n = 2). Thus, 96% (173/180) of the expected number of data sets could be included in the final data set (Faros: n = 39, Tcore: n = 39, GPS: n = 39, Hygrometer: n = 17, GENEActiv: n = 39). Fig 3 shows the data completeness (i.e., useable data points per recording) of individual recordings across recording devices and sex.

Despite everyday activities (e.g., homework, fieldwork and childcare) and varying ambient conditions, the average data completeness was relatively high for recordings with Faros, Tcore, Hygrometer, and GENEActiv (all ≥ 92%). The use of

**Table 1. Characteristics of study participants.**

|  | All (N = 40) | Women (n = 20) | Men (n = 20) | p-value |
|---|---|---|---|---|
| **Age (years)** | 33.7 ± 9.6 | 31 ± 9.4 | 36.4 ± 9.2 | .072[a] |
| **Weight (kg)** | 65.2 ± 10.6 | 61 ± 8.2 | 69.7 ± 11.1 | .007[b] |
| **Height (cm)** | 167.5 ± 11 | 163.2 ± 6 | 171.8 ± 13.3 | <.001[b] |
| **SBP (mmHg)** | 119 ± 15 | 115 ± 12 | 122 ± 16 | .120[a] |
| **DBP (mmHg)** | 77 ± 12 | 73 ± 10 | 80 ± 14 | .088[a] |
| **BMI (kg/m²)** | 23.3 ± 3.5 | 22.83 ± 2.76 | 23.75 ± 4.13 | .818[b] |
| **TBW (L)** | 34.4 ± 6 | 30.6 ± 2.7 | 39.9 ± 4.9 | <.001[b] |
| **FM (%)** | 21.8 ± 8.2 | 22.3 ± 6.6 | 21 ± 10.4 | .286[b] |

Data are M ± SD.

SBP, systolic blood pressure; DBP, diastolic blood pressure; BMI, body mass index; TBW, total body water; FM, fat mass.

[a]calculated using unpaired, two-sided t-test.

[b]calculated using Wilcoxon rank-sum test (two-sided).

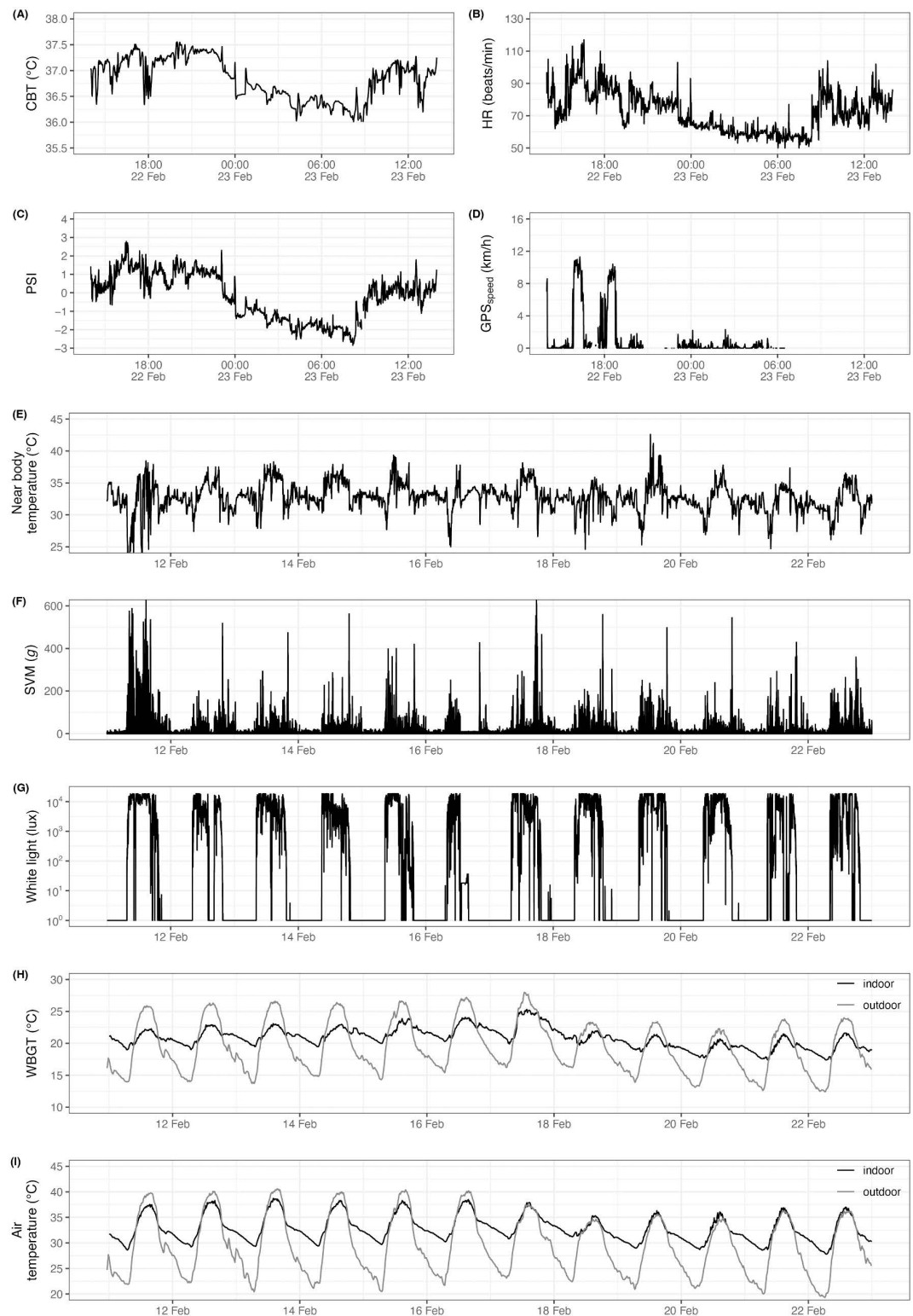

**Fig 2. Physiological and environmental recording of a male participant.** 24-hour time series of his core body temperature (A), heart rate (B), physical strain index (C) and speed (D). 12-day time series of his near body temperature (E) and sum of signal vector magnitude of the tri-axial accelerometer signal (F). Additionally, 12-day time series of his light exposure (G), as well as wet bulb globe temperature (H) and air temperature (I) indoor (at home) and outdoor (weather station in the area).

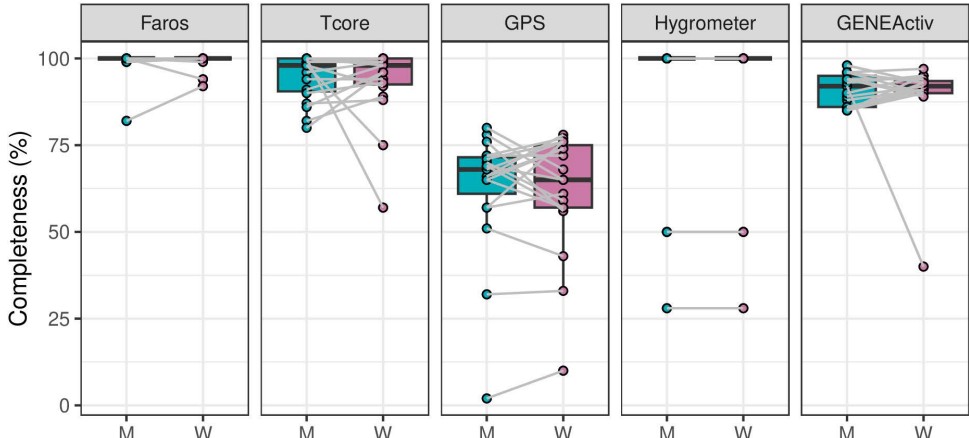

**Fig 3. Completeness of data across devices.** Distribution of individual-level percentages of data recordings across various devices by boxplots and individual data points. A grey line indicates two individuals from the same household. 24-hour electrocardiography (Faros), core body temperature (Tcore) and global positioning system (GPS) recordings, as well as 12-day recordings of indoor wet bulb globe temperature (Hygrometer) and actimetry (GENEActiv).

GPS appears to be less successful, with an average data completeness of only 67%. However, the completeness of the GPS data for a measurement period of 24 hours, as in the present study, can only be a maximum of 70 to 83%, since the operating time of this GPS device is only 17–20 hours according to the manufacturer.

As indicated in Fig 3 data completeness were comparable across sex for Faros, Tcore, GPS, and GENEActiv device (all $p ≥ .755$).

Interestingly, the majority of the available GENEActiv recordings (36/39) were 100% complete (i.e., no data loss during continuous recording for 12 full days) before the non-wear detection algorithms was applied. After use, non-wear episodes were detected in all n = 36 recordings. The completeness of the individual records decreased by an average of 8% (Mdn: 7.5%, IQR 5.8-10.0%). With 12 full days of recording per participant, this corresponds to almost 2 hours per day during which the GENEActiv was apparently removed from the wrist. In addition, on only 15% of the total days of recording (67/432), the GENEActiv appeared to be continuously attached to the wrist.

PSI values could be derived from n = 38 24-hour time series of CBT and HR. Fig 4 shows that the PSI follows a daily rhythm.

For women and men, the PSI was lowest in the morning (6:00) and highest in the evening (19:00). The PSI tended to rise earlier from the morning trough in women than in men. In addition, the women's PSI seems to be higher than men's from morning to evening. However, the average 24-hour PSI profile ranged between −1 and 1.5 for women and men. The histogram of all PSI shows that some individuals experienced a lower or higher level of physical strain. (Fig 4B).

## Acceptability

All 40 administered questionnaires were completed. Neither local study personnel nor participants reported any specific problems regarding the computer-assisted personal interview method. The data was complete and retrieved from our study server without further issues.

Fig 5 shows the percentages of agreement towards specific statements describing the subject's experience. Positive statements were defined as statements connecting positive attributes to the wearables or the experience of wearing them. On average, positive statements had an 89% agreement (79–95%), with participants either agreeing or strongly agreeing. Men expressed an average agreement of 95% and women of 83%. Negative statements were defined as statements

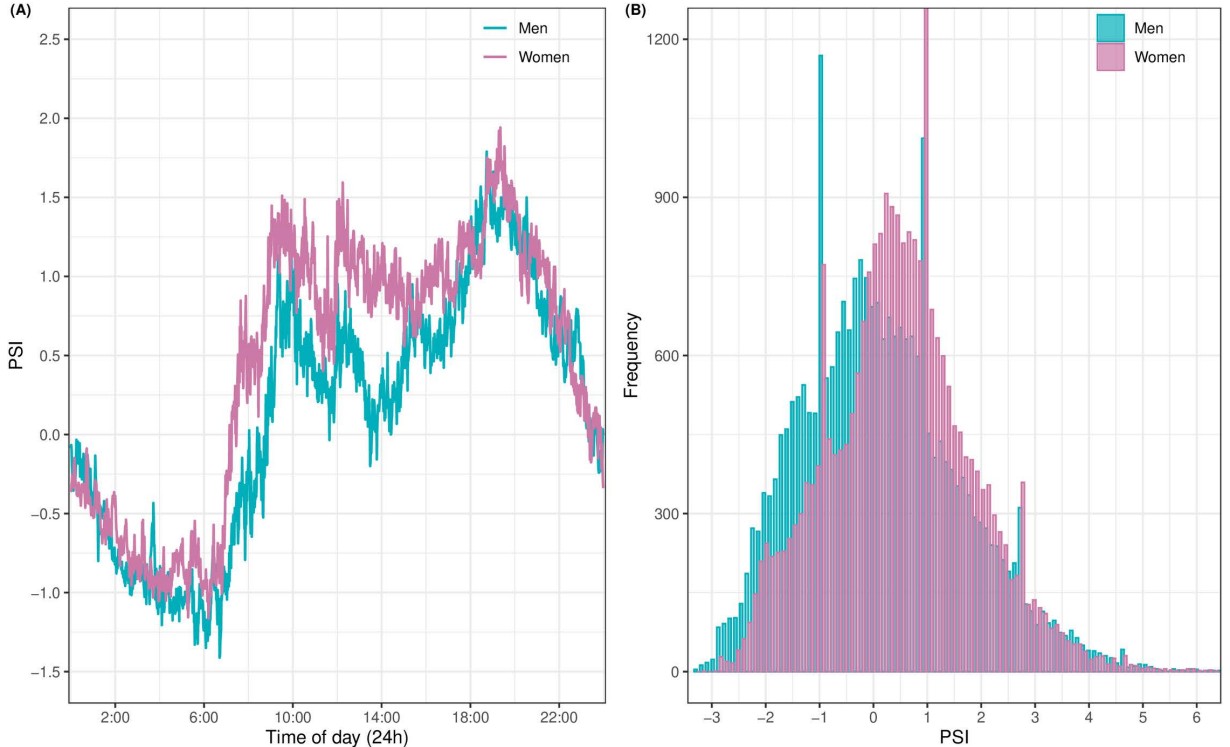

**Fig 4. PSI scores from n = 38 24-hour recordings.** Average PSI profiles (A) and a histogram (B) of PSI from n = 19 women and n = 19 men.

attributing negative characteristics to the wearables or the experience of wearing them. Negative statements had on average, a 35% agreement (2–95%), with men expressing an average agreement of 40% and women of 36%.

In 18% of cases, participants expressed that the wearables affected their working day in some shape or form, with 8% average agreement among men and 27% among women. However, only a maximum of 5% of men either experienced the need to interrupt their routine or a limitation of movement. This amounted to 1% and 4% for women, respectively (Fig 5). Further, 90% of participants did not experience the sensors as overly intrusive in their daily routine. When asked about specific adverse reactions, the most common ones were the need to check the sensor frequently, especially for the GENEActiv (72.5%, n = 29), and the disturbance of personal hygiene and daily care, particularly for the Faros (37.5%, n = 15) and the Tcore (30%, n = 12).

Discrepancies between men and women were most substantial for the perceived lack of sense in wearing the devices, as only 25% of men but 70% of women agreed with such a statement ("I found that wearing the device did not make sense"). Another apparent difference was the perceived strangeness of using the device, where 48% of men and 93% of women agreed with said statement ("I found the device strange to wear/use"). Table 2 depicts the statistical differences in women versus men for 9 exemplarily chosen statements.

The most positively received sensor was the Tcore with an average agreement of 93% towards positive statements. The second most positively received sensor was the GENEActiv with a 92% average agreement rate. The Faros was received most favorable among women averaging out at 99% regarding positive statements. Men received the GENEActiv most positively, averaging 93% in agreeance towards positive statements.

The most negatively received sensor was the Faros with 39% average agreement towards negative statements. Especially strong agreement was recorded towards the statement, that the sensor required too much attention. For women, the

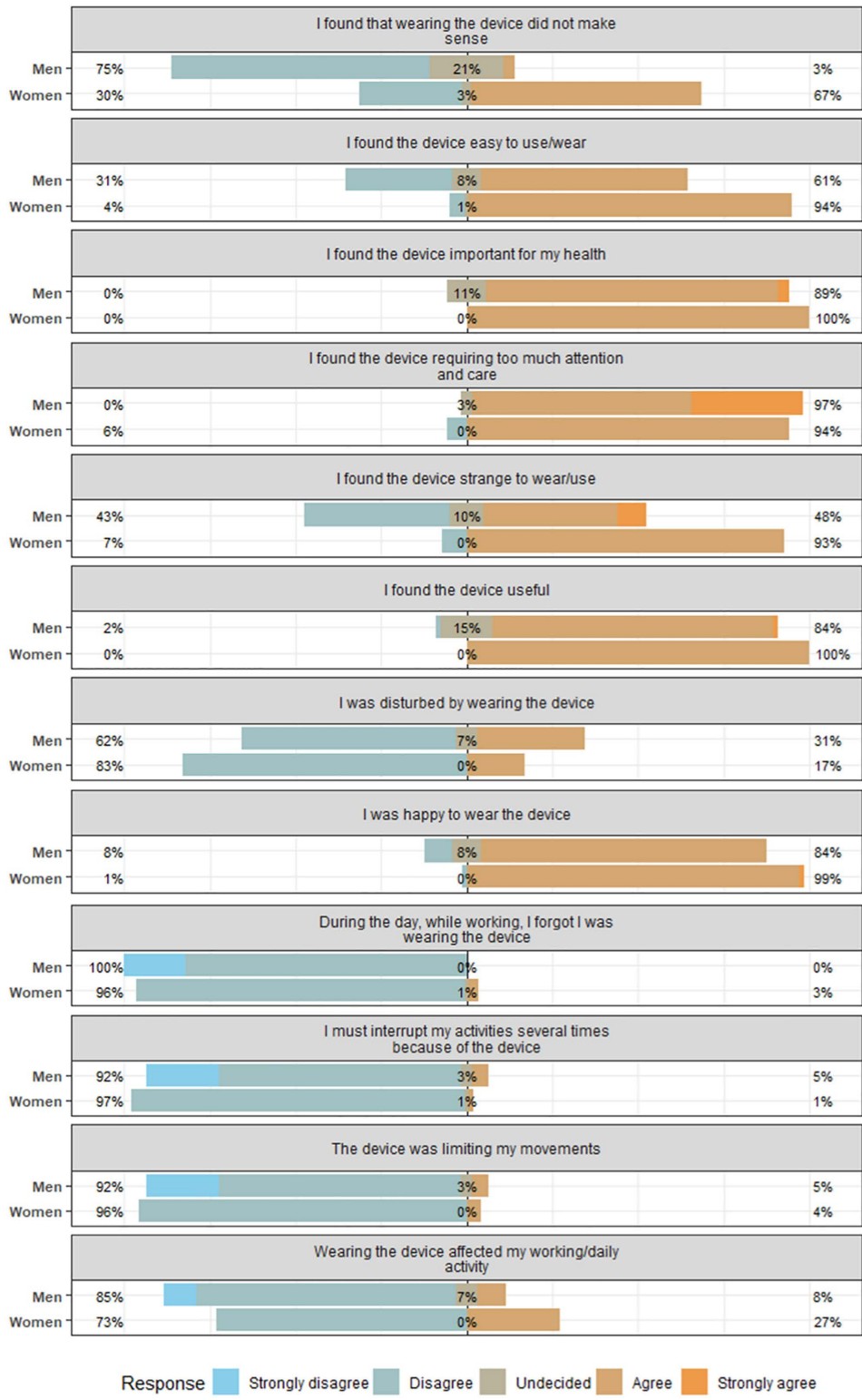

**Fig 5. Responses to Likert-type items grouped for women and men.** The percentages at the edges of the bars are totals of all agreement or disagreement. Due to the rounding of percentages for visual clarity, the summed percentages across categories for each Likert item may not exactly equal 100% (±1%). This rounding does not affect the interpretation of the data.

**Table 2. Statements about the participants' experience in dealing with the wearables.**

| Statement | Women | Men | p-value[a] |
|---|---|---|---|
| I was happy to wear the sensor | 2.03±.27 | 2.2±.54 | .006 |
| I found the sensor easy to wear | 2.14±.47 | 2.55±86 | <.001 |
| I found the sensor useful | 2.04±.19 | 2.14±.41 | .056 |
| I found the sensor important for my health | 2.04±.19 | 2.08±35 | .375 |
| I was disturbed by wearing the sensor | 3.64±.78 | 3.34±.91 | .021 |
| I found the sensor difficult to wear | 3.86±.44 | 3.34±.9 | <.001 |
| I found the sensor strange to wear | 2.25±.63 | 2.99±1.04 | <.001 |
| I found the sensor requiring too much attention | 2.11±.48 | 1.81±.55 | <.001 |
| I must interrupt my working activities | 3.95±.27 | 4.08±.61 | .012 |

Data are M±SD.

[a]calculated using the Wilcoxon rank-sum test (two-sided).

most negatively received sensors were the Faros and GPS data logger both with an average of 36% agreement towards negative statements. For men it was the Faros averaging at 43%. Overall, the least negatively received sensor was the GENEActiv at an average agreement rate of 24%. Moreover, it was the least negatively received sensor for both men and women.

Overall, participants reported no specific reactions from their peers to the wearables (75–85%). There were two notable differences between men and women: (I) when asked to elaborate whether they were comfortable wearing the sensor in public, an average of 46% of men said they were, with 25–30% reporting being comfortable wearing the Faros, the Tcore, and the GPS sensor and 100% of men being comfortable wearing the GENEActiv. On the contrary, 94% of women reported being comfortable wearing the sensors in public, with all sensors receiving 90% or higher approval. (II) on average, 36% of men were asked by their peers about the sensors they wore, with an interest in specific wearables ranging 30–45%. However, on average, only 13% of women were asked about the sensors, with interest focused on the Tcore (15%) and the GENEActiv (10%).

We evaluated hindering factors for further or repeated participation in a similar study setting by having participants answer closed questions, as shown in Table 3. The most prevalent hindering factors were the lack of desired sensor feedback for 70%, the required interaction level for 67.5%, and the length of time required to wear for 52.5% of participants (Table 3).

**Table 3. Closed yes-no questions examining hindering factors for further participation in such a study setup (N = 40).**

| | Yes% (qty) | No% (qty) |
|---|---|---|
| Length of time required to wear | 52.5% (21) | 47.5% (19) |
| Interaction level required | 67.5% (27) | 32.5% (13) |
| Lack of desired sensor feedback | 70% (28) | 30% (12) |
| Adverse effects | 15% (6) | 85% (34) |
| Social acceptance | 15% (6) | 85% (34) |
| Disturbance of daily activity | 2.5% (1) | 97.5% (39) |
| Disturbance of sleep | 5% (2) | 95% (38) |
| Disturbance of personal hygiene | 32.5% (13) | 67.5% (27) |

Qty, number of recordings as absolute values.

## Discussion

Our observations represent novel documented evidence of the technical feasibility and social acceptability of research-grade wearables for continuously monitoring population groups in challenging, naturalistic environments, such as subsistence farmers in rural Burkina Faso. (I) An average of 97.5% of expected data sets were retrieved for all wearables, 85% for indoor WBGT sensors; (II) Individual data point completeness of > 92% for Tcore, Faros, GENEActiv and hygrometer data and an average of 67% for GPS data was achieved; (III) Positive participant feedback averaged 83%. These results underscore the reliability of our approach. This complements the literature on mobile health technologies, emphasizing their advancements and pivotal role in remote health monitoring and directly addressing the lack of such engagement in low-resource contexts [56]. Hereby, we also expand the platform for future long-term, in-depth investigations of heat stress and occupational health in real-world scenarios, enabling the integration of existing methodology with a novel approach in an uninvestigated, low-resource setting of SSA [47,48]. The positive resonance in our study sample attests to the wearables' acceptability in this population, which aligns with the findings from studies in similar contexts on the use of wearable health technologies [57,58]. In contrast, the lack of sensor feedback, the unclear meaning of the wearables, and the perceived time and attention they required were the participants' main adverse experiences.

### Feasibility

Feasibility studies are critical in developing novel approaches and should demonstrate the ability to implement interventions successfully [59]. Within our predefined framework, we consider an overall data set coverage of at least 92% for all employed wearables and data quality reduced by a maximum of 8.3%, a successful demonstration of the technical realizability of our approach. As Cho et al. pointed out, key factors potentially compromising data quality and completeness are sensor and administration-related issues, user compliance, and data synchronization [60]. Further, operating a multitude of complex research-grade wearables requires specific knowledge. Due to different data formats and multiple software requirements, the tasks of data saving, transfer, synchronization, and post-processing required sufficient time, adequate infrastructure, and expertise. To address these pitfalls, we prepared our study setup accordingly. Firstly, we made sure on-site operators were trained and accustomed to the devices, ensuring a proper administration. Sequentially, this also applied to researchers dealing with data processing and analysis. Secondly, we selected pairs of participants living in the same household. This reduced the dropout risk, and heightened the probability of protocol compliance, i.e., continuous and instruction-conform wear of wearables, thus contributing to the observed high data completeness and data quality. Finally, we based the framework of our study on selecting research-grade, established wearables. Monitoring a similar rural population in Burkina Faso employing consumer-grade wearables, Huhn et al. referred missing data in their study to technical difficulties, environmental strain, or prolonged physical stress, potentially impacting the proper positioning or fixation of a device [21]. Notably, in the research of Matzke et al., implementing a similar thermo-patch as Huhn et al., significantly less data was missing, which was attributed to an enhanced adhesion [20]. Here, the two-fold fixation of the Tcore, among other factors, might have contributed to the low data point missingness we observed.

The dual heat-flux method of the Tcore CBT sensor, is considered a relatively novel way to monitor CBT. As it is non-invasive, it cannot measure the actual internal core temperature of the body the way an esophageal catheter or a rectal probe can. However, it has been validated multiple times against rectal thermistors or rectal temperature probes both in clinical and non-clinical settings [61,62]. It has shown to be a convenient method for prolonged continuous CBT monitoring in harsh environments, such as outer space, and was proven to be sufficiently accurate under exercise heat-stress conditions [36,62]. A major advantage is the non-invasive setup, which is of great importance in our chosen setting, as it provides higher comfort for participants and thus minimizes drop-out rates significantly, especially for long-term investigations. As we conduct non-clinical, real-world research the ability to observe physiological responses to environmental stressors and internal heat load changes while maintaining the naturalistic setting, i.e. no changes in daily routine and work due to discomfort is of paramour importance.

The GENEActiv accelerometer has been previously validated regarding its technical reliability against a mechanical shaker and for its estimation of physical activity against VO2 as the criterion measure [29]. To demonstrate its feasibility in the context of this research work, we focused on the completeness and validity of the accelerometric raw data it generated. Despite showing very high data completeness and validity, there are several aspects to consider. As such, the anatomical positioning of a wearable and specifically an accelerometer can significantly affect the generated data, with for example acceleration from wrist-worn accelerometers being generally higher and containing more noise, than from hip-mounted ones [63,64]. To reduce possible noise, we positioned the GENEActiv on the non-dominant wrist of participants. Further, the reproducibility of data may be impaired, if wearables are removed and reattached by non-trained persons, a scenario, that may occur during prolonged monitoring in a free-living setting such the one presented in this study [65]. Our recruitment strategy of couples following the same study protocol to ensure higher compliance, might have alleviated this problem. We focused on demonstrating the technical realizability of prolonged accelerometric monitoring in the given environment. However, the generated ACT data owes great potential for further post-processing, with a range of prior scientific work demonstrating the possibilities for deriving data from GENEActiv raw data. For example, the R-based GGIR package enables to derive Euclidean Norm Minus One, sleep period time, sleep efficiency, activity levels clustered by intensity and other data [66]. White et al. developed an approach to derive activity energy expenditure from wrist-worn accelerometry and Durcharme et al. demonstrated the applicability of a validated algorithm for step count estimation [30,34]. Though such calculations go beyond the scope of this feasibility study, they provide an interesting gauge at the potential prolonged ACT monitoring by wrist-worn accelerometry holds. Nonetheless, a future implementation of algorithms and formulas to estimate derived metrics may encounter issues regarding data quality and validity assessment, as sensible cut-off values would need to be established.

Using GPS to precisely track movement and covered distance coupled with ACT, HR and CBT data assessment, is a promising setup for future in-depth investigations of labor capacity. As Rodriguez et al. have demonstrated in their study, GPS can be a useful instrument in complementing the accelerometric measurements to assess physical activity in naturalistic environments [67]. However, the accuracy and reliability of GPS data is not a given, as it can decline due to several factors. For example, if the connection to a satellite is lost, no data is being transmitted [68]. As we observed a certain amount of data missingness, the remote location of our study, might have played into this particular issue. Further, the anatomical placement of the GPS tracker influences the measurements depending on the performed activity, as waist-band mounted trackers tend to overestimate distances during walking but not during cycling [69]. In our study, the GPS tracker was hung around the neck of participants. This placement might explain why we also observed measurements occurring during nightly rest, indicating a certain amount of noise.

The standardized methodology to estimate outdoor WBGT is specified in ISO 7243 [28]. Moreover, the approach developed and published by Liljegren et al. in 2008 is also considered to be an established, precise and still highly relevant method in WBGT estimation [4,70]. In our research, we relied on the method developed by Carter et al. as the weather station we gathered data from, does not have globe thermometers [27]. We acknowledge, that this approach is considered novel and has yet to be validated the same way Liljegren et al.'s approach has been [71]. However, Carter et al. demonstrated that WBGT estimates derived from proxy data, such as wet-bulb temperature are strongly correlated with measurements from commercial WBGT monitors [27]. Therefore, this approach provides a cost-effective and practical alternative for applications like risk mapping and large-scale studies in remote environments, comparable to our setting. Nonetheless, further validation of the method developed by Carter et al. across different environments and larger datasets would indeed be of value. We observed higher average indoor compared to outdoor WBGT throughout both study runs. We cannot rule out that these unusual temperature profiles, where average indoor WBGT > average outdoor WBGT, are necessitated by the estimation methods we used, due to the weather station not having a globe thermometer. However, limited air exchange within the house might also be a crucial factor, enabling heat to build up and remain throughout the day. This explanation is supported by the relatively stable daily indoor

WBGT profile. Additionally, despite average outdoor WBGT being slightly lower, it fluctuated notably and reached significantly higher peaks throughout the day.

Integrating continuous and individualized HR, CBT, ACT measurements into a PSI according coupled with continuous environmental monitoring in real-world settings can directly address the lack of quantitative field studies on health and work capacity investigations in acclimate change scenarios. In our study, we relied on fixed values for $HR_{rest}$ and $CBT_{rest}$ as was done previously in literature, as in the given setting individualized resting values were not feasible to obtain [48]. However, in future investigations a standardized methodology, for example a fixed time of day for baseline resting values, may be an approach to further the precision of individualized PSI calculation in such a real-life setting. Novel approaches, customized to specific environments and circumstances, such as exercise under heat conditions or agricultural work have been recently presented in the literature [72,73]. For example, Buller et al. demonstrated that an incorporation of skin temperature and the gradient of CBT to skin temperature into an adjusted PSI (aPSI) may significantly increase the accuracy of the index and allow for an individualized, more physiologically accurate $CBT_{critical}$ as opposed to a fixed one [72]. By demonstrating, the feasibility of a PSI calculation in our given setting, the aPSI as demonstrated by Buller et al. is worth exploring in future studies adopting our study setup and combining it for example with the use of a skin temperature patch, as was done by Huhn et al. and Matzke et al. in their respective studies [20,21,72].

Additionally, the subject of occupational health could be addressed even further by integrating data, such as working patterns through GPS tracking. For this we present S2 Fig in the supplementary material, which visualizes what a precise tracking of working patterns throughout the day might look like in future studies. Our approach of capturing real-time physiological data, PSI calculation, GPS tracking and WBGT estimation creates a methodological platform, thus bridging a current gap in current research. This investigation offers a robust framework for assessing heat stress's occupational health impacts on Burkina Faso subsistence farmers. Moreover, it will allow future prolonged studies to implement this feasible approach in similar naturalistic scenarios, enabling cross-integration of different data to create more individualized, adaptive indices. Finally, this setup creates a foundation for developing adaptive strategies to reduce heat-related health risks among vulnerable populations in comparable contexts.

## Acceptability

Previous research investigating the acceptance of wearables in low-resource contexts has shown that the local population is often interested in personal health monitoring [58]. This is pivotal in the acceptability of new technologies for health interventions, as perceived lack of use or social rejection of said intervention may pose significant challenges within the scope of the TAM [51]. Our data show high overall agreement (over 75%) with positive statements, indicating a generally positive attitude towards research-grade portable devices. Local communities within our study showed a profound interest in this research, which enabled a successful recruitment process. Our local personnel who carried out the study protocol facilitated this by actively involving community elders and representatives prior to the study, enabling the highest level of transparency. The recruitment of men and women living in the same household contributed to protocol adherence and further social acceptance.

Wearables were received positively, with usefulness and importance for the respective participants' health rated as the leading positive statements, each with over 90% agreement. A self-reporting activity survey showed that men and women in our study sample engaged in gender-specific daily tasks, such as farming or agricultural work for men and household chores for women. Similar gender-specific work distribution was noted by Kieran et al. in their exposé on gender norms in rural Burkina Faso [74]. Therefore, observed gender-specific differences in reactions to wearables may be attributed to differing experiences while performing different tasks. In our sample, for example, farming livestock and fieldwork, both physically demanding tasks, were carried out solely by men, whereas the women primarily attended to work inside or in close proximity of the house. Men were less likely to agree with positive statements about the Faros, perceived it most negatively out of all sensor and experienced is as the wearable interfering the most with their work routine (20%).

Coupled with the behavioral differences, this may indicate that the Faros was perceived as a nuisance during manual labor. Women were affected by wearing the Faros and the Tcore equally, with 35% (n = 7) (for each device) experiencing an impact on their daily work. A possible explanation might be the setup of the Faros including electrodes and cables fixated to the chest, which may be perceived as particularly inconvenient by women. Further, our indoor WBGT estimation showed unusually high values, which may have been caused by heat not properly dissipating throughout the day. As the Tcore is fixated by a sticky patch and a tight headband, it might cause discomfort in a hot environment during work, which might have been perceived as irritating by the women in our study.

Most of the negative statements were broadly disagreed with. The two most striking negative statements gaining high levels of agreement were the perceived attention sensors required (95%) and the unclear meaning attributed to them (72%). Most participants (95%) reported needing to check the GENEActiv constantly. This is likely because it is a wrist-worn sensor and, therefore, overly exposed to environmental stress during labor. Indeed, 67% of participants expressed the interaction time wearables required as a hindrance factor for further participating in similar studies. Aligning with previous research, this might influence participants to be excessively cautious so as not to damage the wearable [75]. The meaninglessness and strangeness of the wearables as experienced by our participants might be rooted in the perceived lack of appropriate sensor feedback – the most prevalent hindrance to further participation in a similar study. As Zanello et al. assessed in their acceptability study on the use of accelerometric devices in a rural setting of Ghana, participants expressed concerns about being unable to see what is recorded [76].

Villagers showed interest in the purpose and usefulness of the sensors worn by our participants, indicating an overall social openness. This is a promising outcome, as social acceptance and positive feedback may also positively influence individual acceptance. An interesting observation was the discrepancy in how often the men (36%) versus the women (13%) in our sample were approached and asked about the wearables by peers. This might also be explained by gender-specific behavioral patterns. As mentioned above, the performed tasks showed women rarely leaving the vicinity of their house, which lowers the chance for possible social interactions.

Considering our findings, future studies should focus on the preparatory and study-accompanying work with the respective communities. Moreover, providing participants with some sort of sensor feedback, i.e., a regular update on their personal physiological data throughout the study, may reduce the feeling of lacking sensor feedback and thus perceived senselessness. However, even without direct sensor feedback, most participants had a positive sentiment toward the research-grade wearables. As such, our study also presents evidence that research-grade wearables can successfully be deployed in real-world environments so that researchers can benefit from their higher precision, longer-lasting batteries, and higher durability compared to consumer-grade wearables.

## Conclusion

This pilot study contributes to the existing literature on research-grade wearables by demonstrating their technical feasibility for prolonged implementation and continuous monitoring in an SSA real-world environment. Despite the technical limitations regarding WBGT estimation and GPS data precision, we provide a solid, manageable framework for future long-term studies looking into large-scale population-based health monitoring in comparable naturalistic environments. A functional and feasible framework is even more important, as research-grade wearables can pose an economic burden on study design and planning, especially when equipping a larger sample of participants. Further, we present this as a platform to incorporate our approach into the adaptation of indices for occupational health monitoring. This is important work considering the need for adaptation strategies to protect population groups disproportionately vulnerable to climate change. Although a less user-friendly interface might lack desired sensor feedback and be perceived as intimidating, the wearables received predominantly positive feedback from participants individually and socially. This bridges a knowledge gap, as it demonstrates acceptance of wearables that are not primarily consumer-orientated in a low-resource context. Further, this shows that subjects in rural Burkina Faso are open to personal health monitoring, adding to the notion of

subjects in comparable contexts being interested in personal health interventions. Our results show that this also applies to interventions based on research-grade technologies. However, our study also points out current challenges in using research-grade wearables, which could be the focus of optimizations in future research. To summarize, this pilot study provides a starting point for further research implementing research-grade technologies to investigate climate changes' physiological and occupational effects in remote, low-resource, real-world environments such as Nouna, Burkina Faso.

## Supporting information

**S1.   Entry Survey. Entry survey for further eligibility.**
(DOCX)

**S2 Fig.   Exemplary individual GPS recording.** The total time spent in the field by the subject was 433 min, the total distance covered within the fields' boundaries was 3339 m. The recording was done on the 23rd of October 2021, between 0 A.M. (00:00:24) and 0 P.M. (23:59:19). During this time span the subject was 4 times within the fields' boundaries.
(TIFF)

**S3.   Acceptability Questionnaire.**
(DOCX)

**S4.   Inclusivity in global research questionnaire.**
(DOCX)

**S5.   Supplement Acceptability. Underlying data.**
(XLSX)

**S6.   Supplement Completeness. Underlying data.**
(XLSX)

**S7.   Supplement PSI. Underlying data.**
(XLSX)

## Acknowledgments

This research work is part of a larger research unit (RU) titled "Climate Change and Health in Sub-Saharan Africa" (FOR 2396). Within the RU, which examines health outcomes related to climate change, employing a multi-perspective and multi-disciplinary approach, we conduct the individual project "Climate change, heat stress and their impact on health and work capacity". We acknowledge that a crucial advantage of this RU frame is the close cooperation with multiple projects within the RU, sharing data, skills, infrastructure, and samples to ensure knowledge transfer and cross-integration of results. We also acknowledge the tremendous effort local CRSN personnel put into recruitment and performing the experimental protocol, which we consider crucial for the success of this research work. We also acknowledge the support from the Deutsche Forschungsgemeinschaft (DFG) and the Open Access Publication Fund of Charité – Universitätsmedizin Berlin for publication costs.

## Author contributions

**Conceptualization:** Hanns-Christian Gunga, Martina Anna Maggioni.

**Data curation:** Stefan Mendt.

**Formal analysis:** Georgi Zout, Sophie Huhn, Stefan Mendt.

**Funding acquisition:** Hanns-Christian Gunga, Martina Anna Maggioni.

**Investigation:** Lucienne Ouermi, Ali Sié, Martina Anna Maggioni.

**Methodology:** Sandra Barteit, Martina Anna Maggioni.

**Project administration:** Martina Anna Maggioni.

**Resources:** Ali Sié, Martina Anna Maggioni.

**Supervision:** Sandra Barteit, Martina Anna Maggioni.

**Validation:** Georgi Zout, Sandra Barteit, Stefan Mendt.

**Visualization:** Georgi Zout, Stefan Mendt.

**Writing – original draft:** Georgi Zout.

**Writing – review & editing:** Georgi Zout, Celine Höver, Edgar Eggert, Hanns-Christian Gunga, Lucienne Ouermi, Ali Sié, Sophie Huhn, Aditi Bunker, Sandra Barteit, Stefan Mendt, Martina Anna Maggioni.

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
