## [Decision Letter · Decision Letter 0]

27 Dec 2024

Dear Dr. Zout,

Thank you for submitting your manuscript to PLOS ONE. After careful consideration, we feel that it has merit but does not fully meet PLOS ONE’s publication criteria as it currently stands. Therefore, we invite you to submit a revised version of the manuscript that addresses the points raised during the review process.

We look forward to receiving your revised manuscript.

Kind regards,

Hidenori Otani, Ph.D.

Academic Editor

PLOS ONE

Journal Requirements: When submitting your revision, we need you to address these additional requirements. 1. Please ensure that your manuscript meets PLOS ONE's style requirements, including those for file naming. The PLOS ONE style templates can be found at https://journals.plos.org/plosone/s/file?id=wjVg/PLOSOne_formatting_sample_main_body.pdf and https://journals.plos.org/plosone/s/file?id=ba62/PLOSOne_formatting_sample_title_authors_affiliations.pdf 2. Please include a complete copy of PLOS’ questionnaire on inclusivity in global research in your revised manuscript. Our policy for research in this area aims to improve transparency in the reporting of research performed outside of researchers’ own country or community. The policy applies to researchers who have travelled to a different country to conduct research, research with Indigenous populations or their lands, and research on cultural artefacts. The questionnaire can also be requested at the journal’s discretion for any other submissions, even if these conditions are not met. Please find more information on the policy and a link to download a blank copy of the questionnaire here: https://journals.plos.org/plosone/s/best-practices-in-research-reporting. Please upload a completed version of your questionnaire as Supporting Information when you resubmit your manuscript. 3. We note that the grant information you provided in the ‘Funding Information’ and ‘Financial Disclosure’ sections do not match.  When you resubmit, please ensure that you provide the correct grant numbers for the awards you received for your study in the ‘Funding Information’ section. 4. Thank you for stating the following financial disclosure: "This study was supported by the German Research Foundation (DFG) as part of the FOR 2936 “Climate Change and Health in Sub-Saharan Africa”. The DFG has not been involved in any research procedures, such as study planning, design, data collection, analysis, or interpretation, nor in writing this research paper or in deciding to submit it for publication. The study has been approved by DFG (FOR 2936/project: 660477)" Please state what role the funders took in the study.  If the funders had no role, please state: ""The funders had no role in study design, data collection and analysis, decision to publish, or preparation of the manuscript."" If this statement is not correct you must amend it as needed. Please include this amended Role of Funder statement in your cover letter; we will change the online submission form on your behalf. 5. In the online submission form, you indicated that "The data underlying the results presented in the study are available upon request to the corresponding author." All PLOS journals now require all data underlying the findings described in their manuscript to be freely available to other researchers, either 1. In a public repository, 2. Within the manuscript itself, or 3. Uploaded as supplementary information.This policy applies to all data except where public deposition would breach compliance with the protocol approved by your research ethics board. If your data cannot be made publicly available for ethical or legal reasons (e.g., public availability would compromise patient privacy), please explain your reasons on resubmission and your exemption request will be escalated for approval. 6. Please include captions for your Supporting Information files at the end of your manuscript, and update any in-text citations to match accordingly. Please see our Supporting Information guidelines for more information: http://journals.plos.org/plosone/s/supporting-information.

Reviewers' comments:

**Comments to the Author**

1. Is the manuscript technically sound, and do the data support the conclusions?

Reviewer #1: Partly

Reviewer #2: No

2. Has the statistical analysis been performed appropriately and rigorously?

Reviewer #1: Yes

Reviewer #2: Yes

3. Have the authors made all data underlying the findings in their manuscript fully available?

Reviewer #1: Yes

Reviewer #2: No

4. Is the manuscript presented in an intelligible fashion and written in standard English?

Reviewer #1: No

Reviewer #2: Yes

Reviewer #1: The purpose of this investigation was to evaluate the feasibility (data completeness and quality) and acceptability of implementing research grade wearable devices in a rural setting of Nouna, Burkina Faso. While this study was very novel, there are several major limitations. First, the study focuses on sex differences in feasibility and acceptability. However, this does not appear the purpose of the study. It is also unclear the purpose of comparing indoor and outdoor WBGT measurements outside of feasibility and acceptability. It is important for the survey to be developed systematically. The authors need to expand on how the survey was developed to ensure the questions align with the study objectives. Below are several items that authors must consider before a deeper evaluation of the manuscript.

Abstract:

Line 28-29: sentence needs to be revised for clarity

Line 39: what does it mean by data was “very”

Line 39: data should be plural consistenly throughout the manuscript

Introduction:

Line 48: has increased by 1.1?

Line 53: The sentence needs to be revised. The term “heat stress” refers to the heat load (i.e., environmental conditions, physical exertion, and PPE). It appears the authors are referring to heat illness in this statement.

Line 109: It is unclear what the authors are referring to by “closing the gap”. Does this mean that the assessment of physiological responses hasn’t been done before? Authors should consider how the collection of physiological responses would benefit workers in SSA.

Authors are encouraged to consider how this information, after collected, will benefit the SAA inhabitants.

Methods

Line 137: what is the rationale for including men and women who are from the same household? Is this referring to the main residence? If not, authors should define the term “household”.

Line 154: author should clarify how many total study visits. Is the first visit part of one of two, study “runs”?

Line 155: make and model of device? What height was the environmental monitor placed at ?

Line 185: it should be noted that it is an estimation of core body temperature (not a direct measure)

The methods for the survey development should be clearly outlined in the methods section.

Data analysis

It is unclear if data analysis is performed on environmental data.

- The variables being collected should be included in the data analysis section (ex. Sleep duration,step count, sleep efficiency) for clarity.

A comparison between sexes is not mentioned before the data analysis section. It should be included previously.

Results

-There is no mention of collection of blood pressure in the methods section.

-Line 247: what was used to determine data quality for WBGT?

- Line 248-251: Authors should consider whether the inclusion of a comparison between indoor and outdoor WBGT should be removed since the overall objective of the study is assessment of data quality and feasibility of implementation.

-Line 260-266: what criteria was used to determine data completeness and quality for GPS and ACT?

- the comparison of men and women does not appear to be an objective of the study.

-304-307: it is unclear whether the thematic subsections were chosen before or after the questionnaire was completed by participants. If it was performed before, the methods should be outlined in more detail and it should be

Reviewer #2: This is a valuable report that discusses the potential of wearable devices in the harsh environment of Africa.

However, there are a number of places throughout the report where the description is insufficient.

In addition, there are some places where the experimental methods and interpretation of the results are not fully explained.

Please consider making revisions, focusing on the points we have commented on.

L.25: You write ‘mean ambient temperatures of 1.5°’, but the unit of temperature should be ‘°C’, not ‘°’.

L.64: You mention‘even moderate heat stress (18° WBGT)’, but WBGT 18°is really ‘moderate heat stress’? In the reference cite, it seems to be mild heat stress.

Also, 18° and 40° are inappropriate units, and should be °C.

L89: Why is PSI not used in this study, even though it is mentioned as an excellent indicator here? If possible, please mention it

L.153-156, L.172-173: Please clearly state the names of the devices used in the main text, not just in Figure S2.

lso, please describe how the portable WBGT meter used here calculates WBGT.

The method specified in ISO 7243 is the only correct method for WBGT, and if a different method is used, it is considered to be nothing more than an estimate, so you need to show in the literature, etc., how the estimate was made. Otherwise, you should indicate that these values are merely estimates.

L170-172: What measurement factors were used to estimate WBGT in this measurement system, and how was it estimated?

Please specify this in the main text so that it is clear to those who have only read this paper, rather than just citing reference 28.

In addition, I have looked at Reference 28, and while there is a certain degree of correlation between the methods for estimating WBGT from meteorological observation data, they are not yet established, and it is thought that more data is needed. It is thought that WBGT is correct when measured using the method specified in ISO 7243, and that the accuracy of WBGT measured using other methods needs to be considered.

L.175: Fig. S3 contains technical information on environmental measurements, but this information is completely inadequate. As mentioned above, please clearly state what measurements were taken and how the WBGT value was calculated (or estimated).

L177-178: You use a wrist-mounted accelerometer, but can you understand the activity status of the whole body with a wrist-mounted accelerometer? I think it would be better to cite any references you have on this point.

L180: Please cite the R package used and also the references for each calculated value.

L183: There is a missing period after ‘during nightly rest [30,31]’.

L185-188: The deep body temperature is measured using a heat flux sensor attached to the forehead, but there is a lack of information about the validity of this. You cite reference 33, but this reference only measures CBT in the same way as this report, and does not demonstrate its validity.

It seems that the paper cited in reference 33 should be cited, not reference 33? Please re-examine the validity of the cited references.

Furthermore, it is our understanding that the estimation of deep body temperature using this method is not yet established, and there is debate about its validity. I think this point should be mentioned in the discussion.

L188-189: Please either provide details of the ‘custom-made headband’ or a list of references.

L193: Please include information that will allow readers to reproduce ‘the native software’.

L197-198: We have checked the WBGT device, and it appears to be a product made by Canmore in Taiwan, not Germany. Please be more precise in your description.

https://asset.conrad.com/media10/add/160267/c1/-/en/001217431ML03/manual-1217431-renkforce-gp-102-g-porter-gps-logger-blue-black.pdf

L214-218, L269-271: The quality of data is evaluated by asking whether ‘all possible data points exist in the valid data set’, but is this enough to determine validity?

If the accuracy of the data is not evaluated, it is difficult to evaluate the quality, isn't it?

Please include your thoughts on this point in your discussion.

L246: The data integrity is said to have reached 85%, but what is the reason for the remaining 15% not being complete?

Please describe this in your consideration to the extent possible

L247-248: You write 'No decline in data quality was observed', but how did you determine that there was no decline in quality?

L249-250: It says that the indoor WBGT was higher than the outdoor WBGT, but normally, the WBGT is thought to be higher outdoors, where the black globe temperature is higher. Please describe in the discussion section any possible reasons why the indoor WBGT was higher this time.

In addition, different methods are used to measure indoor and outdoor WBGT. These differ from the methods specified in ISO 7243, so is it possible to compare indoor and outdoor WBGT in this situation? Please consider this point as well.

L314-318: Did the tendency to make negative statements about receptivity differ depending on the type of wearable device?

Please describe this to the extent possible, as there is a possibility that receptivity will differ depending on the type of device. If you did not obtain any data, please state this in the discussion.

L326-327: The same applies here as above. Were there any differences between the types of device?

**Do you want your identity to be public for this peer review?** For information about this choice, including consent withdrawal, please see our Privacy Policy

Reviewer #1: No

Reviewer #2: No

---

## [Author Response · Author response to Decision Letter 1]

24 Jun 2025

Reviewer comment:

The purpose of this investigation was to evaluate the feasibility (data completeness and quality) and acceptability of implementing research grade wearable devices in a rural setting of Nouna, Burkina Faso. While this study was very novel, there are several major limitations. First, the study focuses on sex differences in feasibility and acceptability. However, this does not appear the purpose of the study. It is also unclear the purpose of comparing indoor and outdoor WBGT measurements outside of feasibility and acceptability. It is important for the survey to be developed systematically. The authors need to expand on how the survey was developed to ensure the questions align with the study objectives. Below are several items that authors must consider before a deeper evaluation of the manuscript.

Authors response:

Thank you for your detailed and constructive feedback. We appreciate your highlighting of the limitations regarding the focus on sex differences and the rationale behind comparing indoor and outdoor WBGT measurements. In response, we have revised the manuscript to clearly delineate our primary objectives related to evaluating feasibility and acceptability. Additionally, we have expanded the section on survey development to explain systematically how the questionnaire was developed. We addressed the raised issues and adjusted our text accordingly as provided in the table above. We believe these improvements enhance both the clarity and scientific rigor of our study, and we thank you once again for your insightful commentary.

Line 28-29: sentence needs to be revised for clarity

Thank you, for the suggestion. We revised the sentence, please see L. 27-29.

Line 39: what does it mean by data was “very”

Thank you for bringing this to our attention, we rewrote this passage. Please see L. 40.

Line 48: has increased by 1.1?

Thank you for your valuable suggestion. We have considered your critique. Our reasoning for phrasing it this way is, that the process of global warming, indicating and increase in temperatures within the word itself, is often referenced in relation to pre-industrial temperature recordings (considered as a sort of baseline). Therefore, we wrote that global warming has “reached 1.1°C relative to pre-industrial levels”.

Line 53: The sentence needs to be revised. The term “heat stress” refers to the heat load (i.e., environmental conditions, physical exertion, and PPE). It appears the authors are referring to heat illness in this statement.

Thank you very much for your suggestion on rephrasing this sentence. We agree with your observation. We rephrased the sentence for precision of language and to express the train of thought more clearly. Please see L. 53-54.

Line 109: It is unclear what the authors are referring to by “closing the gap”. Does this mean that the assessment of physiological responses hasn’t been done before? Authors should consider how the collection of physiological responses would benefit workers in SSA.

Thank you very much for mentioning this. The application and translation of our research to the benefit of the local community and population is a very important and foundational aspect of our work. To emphasize this in our manuscript we adjusted the text accordingly, please refer to L. 108-115.

Authors are encouraged to consider how this information, after collected, will benefit the SAA inhabitants.

Thank you. Please refer to our reply above and to L. 108-118.

Line 137: what is the rationale for including men and women who are from the same household? Is this referring to the main residence? If not, authors should define the term “household”.

Thank you for mentioning this. Yes, household does refer to the main residence. The rationale for including men and women, meaning couples, who share the same household, or main residence, if you will, is to increase protocol adherence, as well as ensure lower drop-out rates and streamline sample planning and recruitment procedures. Further, it allows us to include both men and women to evaluate the feasibility of our approach more thoroughly including a gender-specific perspective. In order to clarify this, we updated the according section of the manuscript, please see L. 144-145.

Line 154: author should clarify how many total study visits. Is the first visit part of one of two, study “runs”?

Thank you for bringing this to our attention. Each study run had a duration of two weeks. There were a total of three contacts with participants during each respective two-week run. The first visit was to the respective participants’ household, where an explanation of procedures, obtainment of informed consent, an entry survey, the handout of a wrist- worn accelerometer, and the positioning of indoor WBGT device were undertaken. The second visit was done by the participants to an assigned health care facility, where they received sensors for 24-h continuous measurements, and underwent Bio-impedance analysis and a medical check-up. Finally, the third visit was again done by the participants to the assigned health care facility at the end of the two-week run, whereupon participants returned the respective wearables and the WBGT sensor and underwent the acceptability questionnaire. We revised the respective section of the manuscript, to further clarify the study procedure for readers. Please refer to L. 158-159.

Line 155: make and model of device? What height was the environmental monitor placed at?

Thank you for bringing this to our attention. We name the make and model of the device later in the text, see L. 190-191. As for the height the monitor was placed at, it was fixated on a wall out of reach from children at a height of about 170 cm. We now added this information to the manuscript, please see L. 191-193.

Line 185: it should be noted that it is an estimation of core body temperature (not a direct measure)

Thank you very much for outlining this. We checked our manuscript for clarity on this subject and refer to the estimation of core body temperature as monitoring, to avoid confusion. To further outline this distinction, we updated our discussion accordingl. There we thoroughly examine the merit of the dual heat-flux method for core body temperature estimation. Please see L. 475-487.

The methods for the survey development should be clearly outlined in the methods section.

We appreciate your instructive critique on this point. The survey was developed jointly by our PI, our co-projects within the research unit and the local study coordinator. It was originally designed in English and later translated to our participants’ native language (French). This is specified in our methods section in L. 227-231. We additionally updated the information in L. 225-228 to better reflect the process of developing our survey.

It is unclear if data analysis is performed on environmental data.

Thank you for bringing this unclarity to our attention. During the process of revising and streamlining our manuscript, we primarily focused on checking environmental data for completeness and removing outliers, if present, that the indoor hygrometer should not be able to measure. Please see L. 267-269, 328-330.

The variables being collected should be included in the data analysis section (ex. Sleep duration,step count, sleep efficiency) for clarity.

Thank you for addressing this issue. During the revision process, we shifted our focus to appropriately demonstrate the primary aim of our study, namely the feasibility of equipping a rural population in SSA with wearables for a prolonged time. We adjusted our data processing and analysis, accordingly, now evaluating data completeness and, if possible, data quality. Therefore, we decided to not include derived calculated data such as sleep duration or step count into our analysis, as it went beyond the scope of this investigation. Instead, we focused on presenting the raw accelerometric data and exploring its completeness/the validity of measurement.

A comparison between sexes is not mentioned before the data analysis section. It should be included previously

Thank you for this suggestion. We intended to primarily show the technical feasibility and social acceptability of our approach and setup. However, we acknowledge, that perception and acceptability vary between genders, most probably due to behavioral factors. To specify this, we added accordingly to our results section, portraying gender-specific differences we assessed through our acceptability questionnaire. Additionally, we provide a comparison for anthropometric data. To not confuse readers, we now included a specification, that we intend to show some comparisons between men and women in the data analysis section of the methods. Please see L. 280-282.

There is no mention of collection of blood pressure in the methods section.

Thank you for mentioning this. The blood pressure measurement was part of the medical check-up and the successive assessment of anthropometric data. However, to clarify this for readers, we have updated this part of the methods section and included the specific measurements taken during medical check-up. Please see L. 172-174.

Line 247: what was used to determine data quality for WBGT?

Thank you for inquiring this. To ensure quality we relied on previously developed methodology. We directly measured for indoor WBGT implementing a portable hygrometer, which operates according to the method specified in ISO 7234 (WBGT without solar load = 0.7 × Tₙwᵦ + 0.3 × Tg). We estimated outdoor WBGT by gathering data from an automated weather station and applying the method developed by Carter et al. We acknowledge that the calculation based on the work of Carter et al. is a novel approach, that requires further validation and application to heighten its reliability. However, it still constitutes a feasible approach in our given setting. For additional clarity, we now address this in our revised discussion. please see L. 527-546.

Line 248-251: Authors should consider whether the inclusion of a comparison between indoor and outdoor WBGT should be removed since the overall objective of the study is assessment of data quality and feasibility of implementation.

Thank you very much for pointing this out, as it was a significant input for adjusting and refining our work. After consideration and revision, we decided, that to focus our work more on the primary goal of exploring the technical feasibility of our approach, we will exclude any comparisons that are beyond its scope. That is why, excluded comparison of WBGT.

Line 260-266: what criteria was used to determine data completeness and quality for GPS and ACT?

We appreciate your inquiry, as it helps to refine and adjust our manuscript. We acknowledge a lack of clarity in the previous version of our manuscript. To address this, we adjusted and redid parts of our data analysis and processing, focusing more on factors relevant for the primary objective, the technical feasibility of employing wearables for prolonged monitoring in rural SSA. Therefore, ACT data, which is a continuous accelerometric data stream and not directly physiological data with established cut-off values (for excluding outliers), was solely analyzed for data completeness. Here we decided to declare data points valid/complete, if the GENEActiv has truly been worn during measurement, which we determined by analyzing the near-body temperature it records. We applied the same reasoning to GPS data, focusing primarily on data completeness, as it is also not physiological data with established cut-off values. For our adjusted methodology please see L. 257-267.

the comparison of men and women does not appear to be an objective of the study

Thank you for pointing this unclarity out. Please see our response above.

304-307: it is unclear whether the thematic subsections were chosen before or after the questionnaire was completed by participants. If it was performed before, the methods should be outlined in more detail and it should be

Thank you for the instructive feedback. Indeed, the questionnaire was designed in advance by the PI and the local study coordinator. It was based on previous experience and the knowledge of the community. We grouped the questionnaire into fixed thematic subsections after administering it. We took your suggestions into consideration and specified this in the updated text, please refer to L. 240-244.

Reviewer comment:

This is a valuable report that discusses the potential of wearable devices in the harsh environment of Africa.

However, there are a number of places throughout the report where the description is insufficient.

In addition, there are some places where the experimental methods and interpretation of the results are not fully explained.

Please consider making revisions, focusing on the points we have commented on.

Authors response:

We thank you for your insightful and constructive commentary, which has been invaluable in refining our work. We acknowledge that some areas of our report required more detailed descriptions and clearer explanations of our experimental methods and results interpretation. In response, we have thoroughly revised these sections to provide additional context and clarity, ensuring that our methodology and findings are more comprehensively presented and reproducible. We addressed the raised issues and adjusted the manuscript processing accordingly as presented in detail in the table above. Our revisions include expanded details on the data collection process, adjustments in data processing and analysis, and a more rigorous discussion of our methodology and the implications of our results. We are confident that these improvements enhance the overall clarity and scientific rigor of our report. Thank you again for your thoughtful feedback, which has significantly contributed to the revision process and general quality of our manuscript.

L.25: You write ‘mean ambient temperatures of 1.5°’, but the unit of temperature should be ‘°C’, not ‘°’.

Thank you for pointing this out, we have adapted this accordingly.

L.64: You mention‘even moderate heat stress (18° WBGT)’, but WBGT 18°is really ‘moderate heat stress’? In the reference cite, it seems to be mild heat stress.Also, 18° and 40° are inappropriate units, and should be °C.

Thank you very much for bringing this to our attention. We agree that the terminology must be adjusted accordingly and have done so in the manuscript. Please see L. 65. To clarify, we chose the reference in question as the authors present novel insights on heat stress, especially so-called mild heat stress, which was formerly considered to have at best minimal effects on physical work capacity. The authors juxtapose their findings to established thresholds previously published, for example by Kjellström et al., and argue that heat stress even at as low as 18°C WGBT has been previously underestimated regarding its effect on the reduction of physical work capacity.

L89: Why is PSI not used in this study, even though it is mentioned as an excellent indicator here? If possible, please mention it

Thank you very much for this suggestion. We do agree that this is an important consideration. Our overarching goal is to create a platform for future investigations into occupational heat stress, including the adaption of indices such as PSI to ultimately enable the development of preventive and adaptive strategies. To demonstrate the feasibility and suitability of our methodological approach for such a goal, we added the calculation of PSI to our manuscript. We based the calculations on the works of Davey et al. and Byrne et al. Please see L. 253-257.

L.153-156, L.172-173: Please clearly state the names of the devices used in the main text, not just in Figure

Thank you for addressing this. The names of the implemented devices are cited later in the methods section of the manuscript within the sections dedicated to monitoring. We indeed checked once again on the completeness of the stated information and if needed adjusted it accordingly. Please see L. 200, 207-207, 213, 218.

lso, please describe how the portable WBGT mete

---

## [Decision Letter · Decision Letter 1]

7 Aug 2025

Feasibility and acceptability of research-grade wearables for health and labor capacity monitoring in the context of climate change and heat stress: the case of Nouna, Burkina Faso

PONE-D-24-28297R1

Dear Dr. Zout,

We’re pleased to inform you that your manuscript has been judged scientifically suitable for publication and will be formally accepted for publication once it meets all outstanding technical requirements.

Kind regards,

Hidenori Otani, Ph.D.

Academic Editor

PLOS ONE

Reviewers' comments:

Reviewer's Responses to Questions

**Comments to the Author**

Reviewer #1: All comments have been addressed

Reviewer #2: All comments have been addressed

2. Is the manuscript technically sound, and do the data support the conclusions?

Reviewer #1: (No Response)

Reviewer #2: Yes

3. Has the statistical analysis been performed appropriately and rigorously?

Reviewer #1: (No Response)

Reviewer #2: Yes

4. Have the authors made all data underlying the findings in their manuscript fully available?

Reviewer #1: (No Response)

Reviewer #2: Yes

5. Is the manuscript presented in an intelligible fashion and written in standard English?

Reviewer #1: (No Response)

Reviewer #2: Yes

Reviewer #1: (No Response)

Reviewer #2: Thank you for reviewing the revised manuscript.

I have confirmed that you have responded appropriately to the review comments. Therefore, I have decided to accept this manuscrip

**Do you want your identity to be public for this peer review?** For information about this choice, including consent withdrawal, please see our Privacy Policy

Reviewer #1: No

Reviewer #2: No

---

## [Editor Report · Acceptance letter]

PONE-D-24-28297R1

PLOS ONE

Dear Dr. Zout,

I'm pleased to inform you that your manuscript has been deemed suitable for publication in PLOS ONE. Congratulations! Your manuscript is now being handed over to our production team.

Kind regards,

on behalf of

Dr. Hidenori Otani

Academic Editor

PLOS ONE